# Endurance exercise ameliorates phenotypes in *Drosophila* models of spinocerebellar ataxias

**Alyson Sujkowski[1,2], Kristin Richardson[1], Matthew V Prifti[2], Robert J Wessells[1]\*, Sokol V Todi[2,3]\***

[1]Department of Physiology, Wayne State University School of Medicine, Detroit, United States; [2]Department of Pharmacology, Wayne State University School of Medicine, Detroit, United States; [3]Department of Neurology, Wayne State University School of Medicine, Detroit, United States

**\*For correspondence:**
rwessell@med.wayne.edu (RJW);
stodi@wayne.edu (SVT)

**Competing interest:** The authors declare that no competing interests exist.

**Abstract** Endurance exercise is a potent intervention with widespread benefits proven to reduce disease incidence and impact across species. While endurance exercise supports neural plasticity, enhanced memory, and reduced neurodegeneration, less is known about the effect of chronic exercise on the progression of movement disorders such as ataxias. Here, we focused on three different types of ataxias, spinocerebellar ataxias type (SCAs) 2, 3, and 6, belonging to the polyglutamine (polyQ) family of neurodegenerative disorders. In *Drosophila* models of these SCAs, flies progressively lose motor function. In this study, we observe marked protection of speed and endurance in exercised SCA2 flies and modest protection in exercised SCA6 models, with no benefit to SCA3 flies. Causative protein levels are reduced in SCA2 flies after chronic exercise, but not in SCA3 models, linking protein levels to exercise-based benefits. Further mechanistic investigation indicates that the exercise-inducible protein, Sestrin (Sesn), suppresses mobility decline and improves early death in SCA2 flies, even without exercise, coincident with disease protein level reduction and increased autophagic flux. These improvements partially depend on previously established functions of Sesn that reduce oxidative damage and modulate mTOR activity. Our study suggests differential responses of polyQ SCAs to exercise, highlighting the potential for more extensive application of exercise-based therapies in the prevention of polyQ neurodegeneration. Defining the mechanisms by which endurance exercise suppresses polyQ SCAs will open the door for more effective treatment for these diseases.

## Editor's evaluation

In this strong study, the investigators examine the therapeutic ability of endurance exercise to disease in *Drosophila* models of the spinocerebellar ataxias types (SCAs) 2, 3, and 6. Results support that exercise induced improvements rely on Sesn and perhaps its role in regulating autophagy.

## Introduction

Nine inherited neurodegenerative disorders are caused by expansion of a CAG triplet repeat in the protein-coding region of the respective disease genes. The CAG repeat encodes the amino acid glutamine; thus, the disease protein carries a lengthened tract of polyglutamine (polyQ) residues that renders it toxic. The polyQ family includes six spinocerebellar ataxias (SCAs 1, 2, 3, 6, 7, and 17), Huntington's disease (HD), dentatorubral-pallidoluysian atrophy, and Kennedy's disease (also known as spinal-bulbar muscular atrophy, or SBMA) (*Paulson et al., 1997*; *Zoghbi and Orr, 2000*; *La Spada*

and Taylor, 2003; Todi et al., 2007; Zoghbi and Orr, 2009; Paulson et al., 2017; Lieberman et al., 2019; Buijsen et al., 2019). Various pathogenic mechanisms are shared, involving primarily toxic gain-of-function acquired by the disease protein. These mutations yield aggregation of the toxic polyQ protein and, despite their wide expression throughout the body, only select central nervous system regions degenerate in each disease. Other common mechanisms underlying polyQ pathogenesis include mitochondrial, transcriptional, and autophagic dysfunction (Sullivan et al., 2019). Although the disorders share disease mechanisms, the individual host proteins have divergent functions (Zoghbi and Orr, 2000; La Spada and Taylor, 2003; Todi et al., 2007; Zoghbi and Orr, 2009; Paulson et al., 2017; Lieberman et al., 2019; Buijsen et al., 2019).

There is an unmet need for the treatment of polyQ diseases as there is currently no therapeutic solution for any of these disorders. Here, we examined the idea that daily endurance exercise might be of benefit against polyQ disorders. Endurance exercise is a low-cost, effective intervention with broad pro-healthspan effects, including reducing the incidence of obesity, heart disease, cancer, and cognitive defects (Booth et al., 2012; Wilmot et al., 2012; Strasser, 2013). Previous studies examined to a limited extent the effects of chronic exercise in patients with advanced stages of ataxia, including polyQ disorders: some types of exercise might help in certain cases (Chang et al., 2015; Schatton et al., 2017; de Oliveira et al., 2018) but not others (D'Abreu et al., 2010; Wang et al., 2018). These studies may have been limited by lack of normalized protocols, genetics, and variable disease stage among patients. Some work on the possible role of exercise was also conducted in mouse models of HD, SBMA, and SCA1. SCA1 and SBMA mice seemed to benefit from exercise (Fryer et al., 2011; Chuang et al., 2019; Chua et al., 2014; Giorgetti et al., 2016; Sujkowski et al., 2022), whereas some models of HD improved (Herbst and Holloway, 2015; Ji et al., 2015) while others did not (Corrochano et al., 2018).

Differences in these investigations could be due to variation in protocols, genetic background, age of testing, and other parameters difficult to normalize among studies. While studies in animal models have identified potential pathways by which exercise may affect polyQ phenotypes, absence of normalization may limit the applicability and extension of those results to other polyQ diseases and further up the evolutionary chain. Here, we take advantage of the *Drosophila* model system to systematically examine which polyQ disorders are responsive to exercise and to identify mechanisms involved.

*Drosophila melanogaster* has been widely and successfully utilized to understand polyQ disease mechanisms (Bonini and Fortini, 2003; Perrimon et al., 2016; Pandey and Rajamma, 2018; Ueyama and Nagai, 2018). In parallel, the fly has emerged as a highly efficient model for studying long-term effects of chronic exercise (Sujkowski and Wessells, 2018). Following a 3 week daily, gradually ramped training program, flies of diverse genotypes experience reproducible changes to speed, endurance, flight, cardiac performance, mitophagy, and mitochondrial function (Piazza et al., 2009; Sujkowski et al., 2012; Laker et al., 2014; Sujkowski et al., 2015; Damschroder et al., 2018b; Sujkowski et al., 2019). These changes track with adaptive physiological changes seen in chronically exercising humans (Sujkowski and Wessells, 2018; Wessells et al., 2018), and induce conserved changes in gene expression, including conserved mediators of exercise benefits, such as peroxisome proliferator-activated receptor γ coactivator 1-α(PGC-1α) and Sestrin (Sesn) (Tinkerhess et al., 2012b; Kim et al., 2020).

Here, we tested the effects of chronic exercise in three *Drosophila* models of polyQ SCAs: 2, 3, and 6. Ectopic expression of these human, polyQ-expanded disease proteins in flies is toxic (Tsou et al., 2013; Tsou et al., 2015a; Tsou et al., 2015b; Costa et al., 2016; Tsou et al., 2016; Sutton et al., 2017; Ristic et al., 2018). We find that exercise has a dramatically positive effect in SCA2 flies, with modest effects on SCA6 model flies, and no benefit to SCA3 flies, indicating a range of response in different disease models. In addition, the exercise-mimicking protein, Sesn, improves early death and rescues mobility declines in SCA2 model flies. We further find that improvement in disease phenotypes depends on functions of Sesn that reduce oxidative damage and modulate the mechanistic target of rapamycin (mTOR). Our observations identify Sesn as a mechanistic target that can be leveraged toward therapeutic options for patients unable to exercise, or to supplement the benefits of those who can.

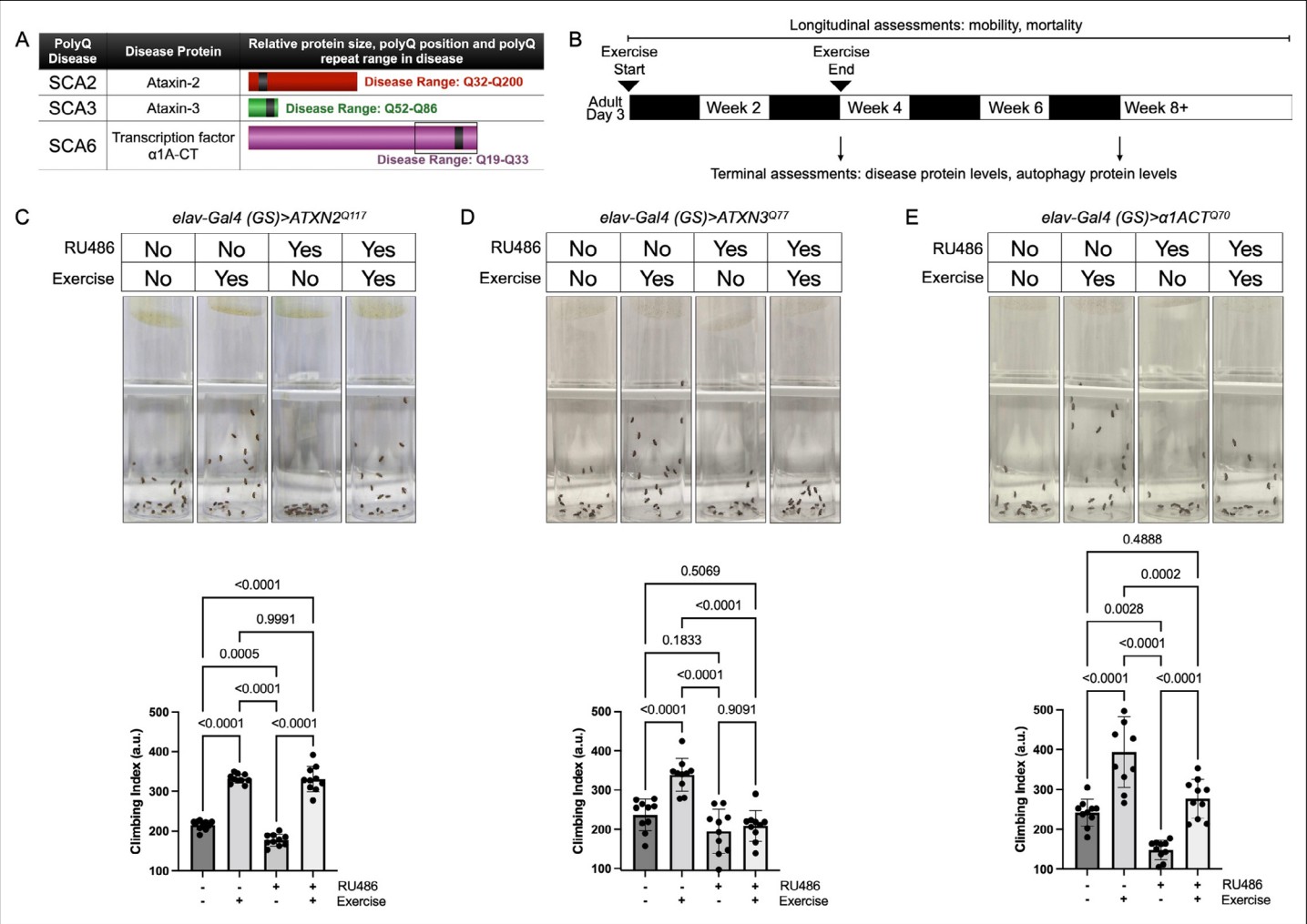

**Figure 1.** Endurance exercise differentially affects mobility in *Drosophila* models of spinocerebellar ataxia. (**A**) Spinocerebellar ataxia (SCA) models used in this study. Box in SCA6 highlights the transcription factor, β1A-CT that is encoded through an internal ribosomal entry site of the β1A transcript. It is this transcript that we utilized for SCA6 model flies. (**B**) Timeline of endurance exercise program and assessment of physiology and disease protein levels. (**C–E**) Representative climbing speed images (upper panels) in *Drosophila* models of (**C**) SCA2, (**D**) SCA3 and (**E**) SCA6. Photos taken 2 s after inducing negative geotaxis response in 4-week-old flies, following endurance exercise completion. Bottom panels: quantification by ANOVA with Tukey's post-hoc for significantly different groups; means ± SD. *n* = 10. Each individual datum depicts average week four climbing speed in arbitrary units (a.u.) for five vials of 20 flies.

The online version of this article includes the following figure supplement(s) for figure 1:

**Figure supplement 1.** Endurance exercise improves climbing speed in uninduced background control flies.

## Results

### Exercise differentially impacts mobility in *Drosophila* polyQ SCA models

Expression of polyQ disease proteins in flies causes reduced motility and neuronal cell death, as we and others have shown in various publications (*Bonini and Fortini, 2003*; *Perrimon et al., 2016*; *Pandey and Rajamma, 2018*; *Ueyama and Nagai, 2018*; *Tsou et al., 2015a*; *Sutton et al., 2017*; *Ristic et al., 2018*; *Jackson et al., 1998*; *Fernandez-Funez et al., 2000*; *Romero et al., 2008*; *Hsu et al., 2014*; *Rosas-Arellano et al., 2018*). To test if endurance exercise reduces polyQ-dependent phenotypes in the fly, we selected three SCAs: 2, 3, and 6 (*Figure 1A*). Full-length, human disease protein was expressed through the binary, RU486-inducible Gal4-UAS system (*Sujkowski et al., 2015*; *Brand et al., 1994*) in all fly neurons and only during adulthood, which is toxic (*Tsou et al., 2013*; *Tsou et al., 2015a*; *Tsou et al., 2015b*; *Costa et al., 2016*; *Tsou et al., 2016*; *Sutton et al., 2017*; *Ristic*

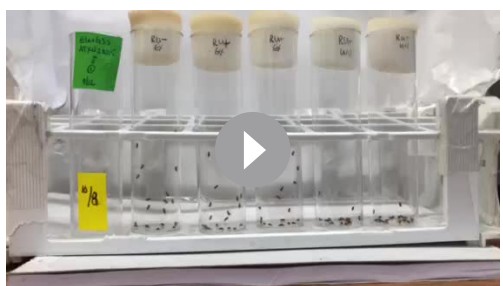

**Video 1.** SCA2 model flies show improved motility with exercise. Video of 4-week-old uninduced control and SCA2 model flies taken after 3 weeks of ramped endurance exercise. Order of vials depicted, left to right: Vial 1-uninduced, exercised control, Vials 2,3-SCA2 model, exercised, Vials 4,5-SCA2 model, unexercised.

https://elifesciences.org/articles/75389/figures#video1

*et al., 2018*). We have established an optimized exercise protocol (*Figure 1B*) that drives reproducible effects that are robust across genotypes (*Piazza et al., 2009*). These effects are not due to non-specific stress or behavioral effect of the regimen—unexercised flies exposed to training, but prevented from running, do not experience improvements (*Piazza et al., 2009*).

Exercised wild-type flies retained faster climbing speed than age-matched, unexercised siblings (*Sujkowski and Wessells, 2018*). SCA2 model flies expressing polyQ-expanded ATXN2 specifically in adult neurons (*elav-Gal4 (GS)> ATXN2^{Q117}*) had reduced climbing speed by 4 weeks of age, but exercise fully rescued climbing speed to the level of age-matched, uninduced control flies that complete the three-week, ramped exercise program (*Figure 1C, Video 1*). In contrast, four-week-old SCA3 model flies (*elav-Gal4 (GS)> ATXN3^{Q77}* RU+) did not increase climbing speed after exercise training (*Figure 1D*). Flies expressing polyQ-expanded β1ACT (*elav-Gal4 (GS)> β1ACTQ^{70}*), the disease gene in SCA6 (*Tsou et al., 2016*), had reduced week-four climbing speed in comparison to uninduced, unexercised control flies, but climbing was partially rescued after exercise training (*Figure 1E*).

Neurodegenerative severity of polyQ disorders tends to increase with age (*Zoghbi and Orr, 2000*; *La Spada and Taylor, 2003*). We have previously shown that exercise-induced improvements to climbing speed and endurance are long-lasting and persist even after the three-week exercise program is complete (*Piazza et al., 2009*). Neither RU486 feeding, nor expression of the empty vector used to generate transgenic flies in adult neurons, altered exercise adaptations to either climbing speed or endurance. Furthermore, expression of an isolated polyQ80 repeat (*Johnson et al., 2020*) in adult neurons reduced climbing speed and abrogated exercise benefits, while age-matched, uninduced control flies adapted to exercise (*Figure 1—figure supplement 1*).

Having confirmed that post-exercise differences in climbing speed can be discerned in polyQ model flies, we next subjected the same three SCA models from *Figure 1* to our endurance exercise program and tested them longitudinally for changes to climbing speed and endurance. In contrast to the ramped exercise program, endurance is tested by placing flies on the climbing apparatus on non-training days and allowing them to climb to exhaustion. When fewer than 20% of flies in a vial no longer respond to the climbing stimulus, the time is recorded, and the vial is removed from the apparatus and scored as fatigued. These data are plotted similarly to a survival curve, with each datum representing a single vial of 20 flies (*Tinkerhess et al., 2012a*).

Spinocerebellar ataxias type 2 model flies had reduced climbing speed by adult week four (*Figure 2A*, compare RU- UN to RU+ UN), but endurance exercise fully rescued climbing speed to the level of exercised, uninduced control flies, and improvement persisted into adult week five (*Figure 2A*, compare RU- EX to RU+ EX). Baseline endurance in SCA2 flies was similar to uninduced, unexercised siblings (log-rank, *P* = 0.2018) and exercised SCA2 flies improved endurance as well as exercised, uninduced controls (*Figure 2B*). We next examined ATXN2 protein levels to see if improved mobility correlated with changes in toxic protein levels. Exercise markedly reduced ATXN2 levels relative to age-matched, unexercised siblings, linking improved physiology to reductions in disease protein (*Figure 2C, I*).

In contrast, SCA3 model flies did not improve climbing speed or endurance (*Figure 2D, E*) after exercise training. Furthermore, exercised and unexercised SCA3 flies had similar levels of ATXN3, the causative, polyQ-expanded disease protein in SCA3 (*D'Abreu et al., 2010*; *Figure 2F, I*). SCA6 model flies had reduced climbing speed at weeks three and five compared to unexercised, uninduced siblings, and exercise training improved climbing speed, albeit not to the level of uninduced, exercised control flies (*Figure 2G*). Unexercised SCA6 flies trended toward lower endurance relative to

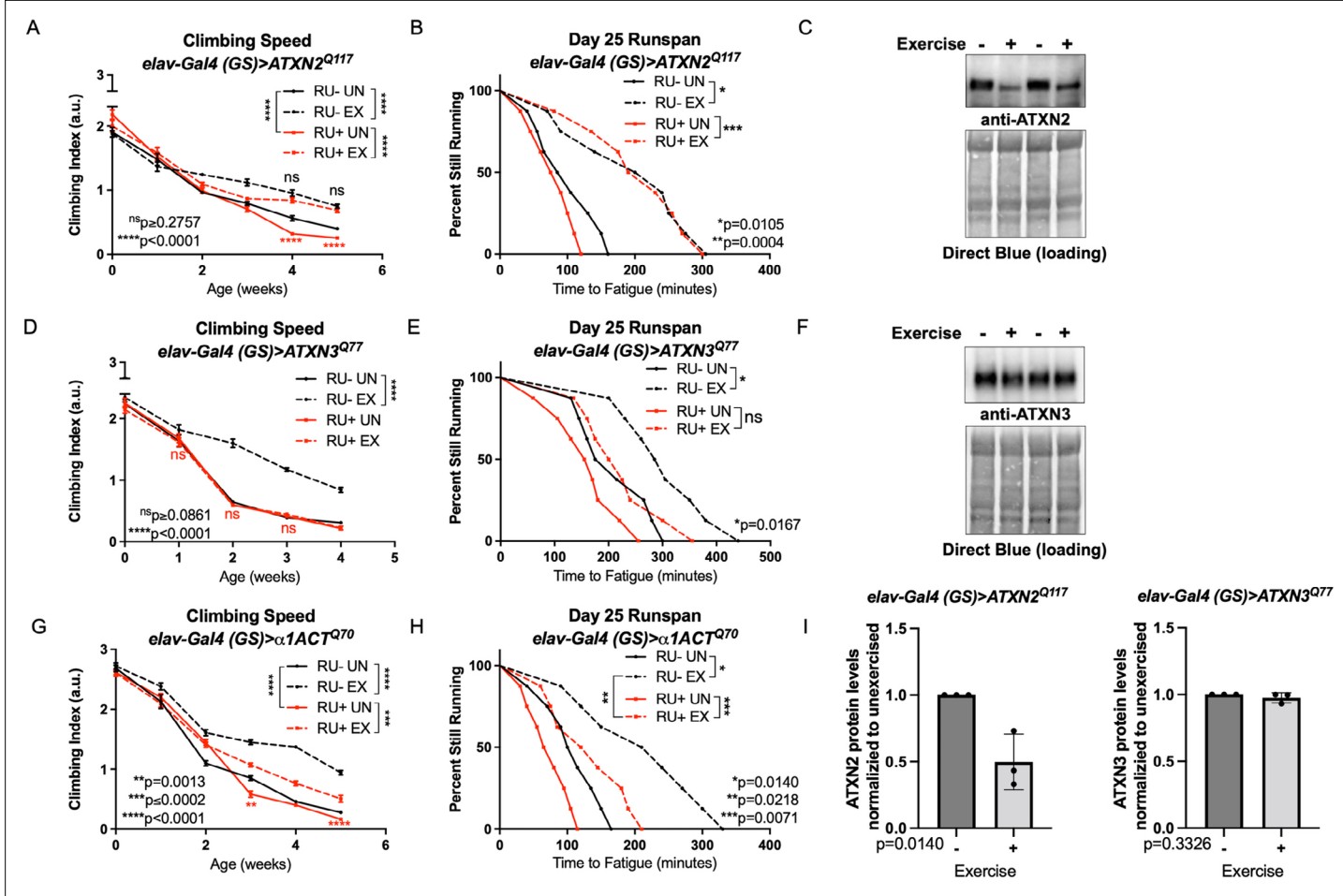

**Figure 2.** Endurance exercise differentially affects mobility and disease protein levels in *Drosophila* models of SCA. (**A**) Flies ectopically expressing polyQ-expanded *ATXN2* in adult neurons (RU+ UN) have lower climbing speed than unexercised, uninduced control flies (RU- UN) by adult week 4. Exercise fully rescues climbing speed to the level of exercised, uninduced control flies (compare RU- EX to RU+ EX). (**B**) Flies expressing polyQ-expanded *ATXN2* in adult neurons (RU+) have similar endurance to uninduced control flies (RU-) whether exercised or not. (**C**) Exercise reduces ATXN2 protein levels in flies ectopically expressing polyQ-expanded *ATXN2* in adult neurons, quantified in (**I**). (**D, E**) Flies ectopically expressing polyQ-expanded *ATXN3* in adult neurons (RU+) have similar (**D**) climbing speed and endurance to uninduced, unexercised control flies and fail to improve either (**D**) climbing speed (**E**) endurance with exercise. (**F**) Exercise does not affect ATXN3 protein levels in flies expressing CAG-expanded *ATXN3* in adult neurons, quantified in (**I**). (**G**) Flies ectopically expressing CAG-expanded *β ACT* in adult neurons (RU+) have lower climbing speed than unexercised, uninduced control flies (RU- UN) by adult week 3, and exercise partially rescues climbing speed, although not to the level of exercised, uninduced control flies. (**H**) Exercise improves endurance in flies expressing CAG-expanded α*1ACT* in adult neurons (RU+ EX), but not to the level of exercised, uninduced control siblings (compare RU- EX to RU+ EX). Mobility and endurance data are presented as representative experiments from triplicate biological replicates. Each individual datum depicts average climbing speed for ≥5 vials of 20 flies, analyzed by 2-way ANOVA (climbing speed), or time for 80% of flies in an individual vial ($n \geq 8$ vials of 20 flies) to reach exhaustion, analyzed by log rank (endurance). Representative Western blots (five flies/lysate) from three biological repetitions, analyzed by ANOVA with Tukey's post-hoc for significantly different groups. Error bars indicate ± SD.

The online version of this article includes the following source data for figure 2:

**Source data 1.** Uncropped, unedited blots from *Figure 2C*.

**Source data 2.** Uncropped, unedited blots from *Figure 2F*.

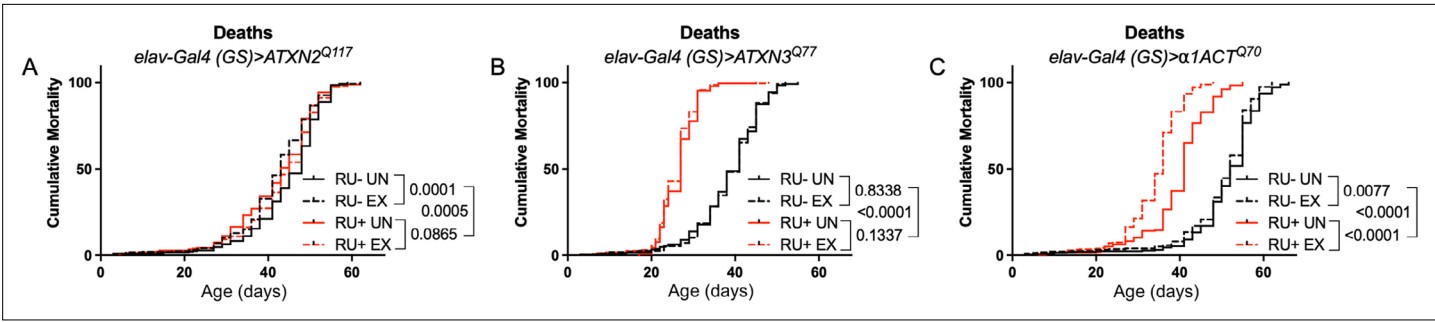

**Figure 3.** Expression of ATXN2 in adult neurons does not negatively impact lifespan. Cumulative mortality in (**A**) *elav-Gal4 (GS)> ATXN2^{Q117}* flies is similar whether exercised or not (compare RU- UN to RU+ UN, and RU- EX to RU+ EX). (**B**) *elav-Gal4 (GS)> ATXN3^{Q77}* flies have reduced lifespan compared to uninduced controls, and exercise does not negatively affect either group. (**C**) *elav-Gal4 (GS)>α1ACT^{Q70}* flies have lower lifespan than uninduced controls, and exercise reduces lifespan further in this cohort. n ≥ 231, analyzed by log-rank tests.

uninduced, unexercised control flies (log-rank, $P = 0.0559$) and improved endurance after exercise training, but did not equal the endurance of uninduced, exercised controls (log-rank, $P = 0.0218$) (*Figure 2H*).

## Sestrin expression improves phenotypes in SCA2 model flies without exercise

We have previously shown that the stress-inducible protein, Sesn, is necessary for exercise adaptations in multiple species (*Kim et al., 2020*), and Sesn activity has been proposed to play a protective role against neurodegeneration (*Chen et al., 2019*). Both endurance exercise and Sesn increase autophagy by inhibiting mTOR (*Sujkowski et al., 2015*; *Mahalakshmi et al., 2020*; *Lee et al., 2010*), and increased autophagy enhances ATXN2 solubility and reduces phenotypes in SCA2 patient cells (*Wardman et al., 2020*). Since muscle-specific dSesn overexpression is sufficient to replicate the beneficial adaptations of endurance exercise, even in sedentary *Drosophila* (*Kim et al., 2020*), we next overexpressed dSesn in SCA2 model flies.

Longevity assessment in SCA model flies expressing polyQ-expanded disease protein selectively in adult neurons confirmed previously reported lifespan reductions in both SCA3 and SCA6 model flies (*Tsou et al., 2016*; *Johnson et al., 2019*; *Figure 3B, C*), but neither ATXN2 expression nor exercise significantly affected longevity in SCA2 flies (*Figure 3A*). CAG expansion in *ATXN2*, a widely expressed gene, causes early mortality in SCA2 patients (*Scoles and Pulst, 2018*). We performed subsequent experiments in flies ubiquitously expressing two copies of polyQ-expanded ATXN2 simultaneously with dSesn overexpression (*sqh> ATXN2^{Q117};dSesn^{WT};ATXN2^{Q117}*) to more closely model clinical subjects. Indeed, constitutive expression of polyQ-expanded ATXN2 in all tissues, throughout development and in adults, (*sqh> ATXN2^{Q117};ATXN2^{Q117}*) caused early death in both female (*Figure 4A*) and male (*Figure 4D*) flies. Mobility reductions appeared earlier than with pan-neuronal expression, present upon adult eclosion from the pupal case and persisting until death (*Figure 4B, E*). On the other hand, flies ubiquitously expressing both ATXN2^{Q117} and dSesn improved survival (*Figure 4A, D*), although rescue was not to the level of background control flies. Mobility improvements were more pronounced, with complete rescue of climbing speed in both females (*Figure 4B*) and males (*Figure 4E*). We next examined disease protein levels in *sqh> ATXN2^{Q117};dSesn^{WT};ATXN2^{Q117}* flies and found significant reduction in ATXN2 compared to SCA2 flies without dSesn expression (*Figure 4C, F*, quantified in Figure 7), similar to aforementioned reduction in disease protein after exercise. In contrast, ubiquitous expression of either ATXN3^{Q77} or β 1ACT^{Q70} led to developmental and early adult lethality (*Tsou et al., 2016*; *Johnson et al., 2019*), and dSesn expression did not improve outcomes (*Figure 4—figure supplement 1*).

## Oxidative resistance and mTOR modulating functions of Sesn are necessary to improve SCA2 phenotypes

Sesn is a multi-functional protein with separate, previously identified oxidoreductase and mTOR interacting domains (*Kim et al., 2015*). The C86S (dSesn^{C86S}) mutation abolishes oxidoreductase activity

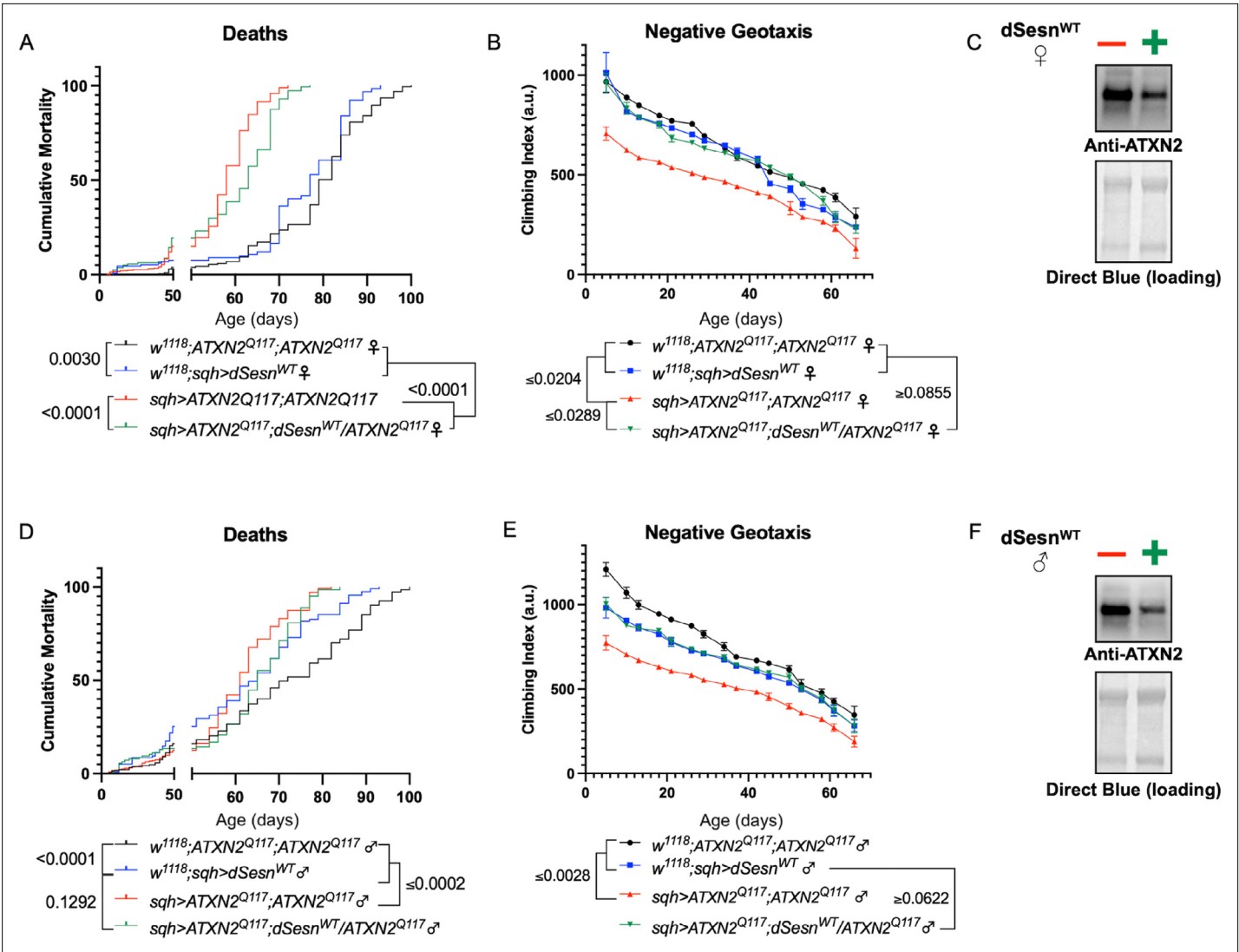

**Figure 4.** dSesn expression improves early death and low mobility in SCA2 flies, concurrent with reduction in disease protein. Female (**A, B**) and male (**D, E**) flies ubiquitously expressing two copies of CAG-expanded *ATXN2* (red lines) have early death (**A, D**) and lower climbing speed (**B, E**) than age-matched background control flies (blue and black lines). dSesn expression in flies also expressing polyQ-expanded ATXN2 (green lines) partially rescues early death (**A, D**) and fully rescues decreased mobility (**B, E**). ATXN2 protein levels are lower in both female (**C**) and male (**F**) flies ubiquitously expressing both dSesn and polyQ-expanded ATXN2. Survival and mobility experiments performed in triplicate. Individual survival experiments comprise ≥200 flies/genotype, scored every second day for death and analyzed by log-rank tests. Each individual climbing datum depicts average climbing speed for ≥5 vials of 20 flies, analyzed by 2-way ANOVA. Error bars indicate ± SD. Representative Western blots from five biological replicates (five flies per lysate), quantified in *Figure 7*.

The online version of this article includes the following source data and figure supplement(s) for figure 4:

**Source data 1.** Uncropped, unedited blots from *Figure 4C*.

**Source data 2.** Uncropped, unedited blots from *Figure 4F*.

**Figure supplement 1.** Ubiquitous dSesn expression fails to increase survival in flies expressing either polyQ-expanded ATXN3 or polyQ-expanded α1ACT.

**Figure supplement 2.** Ubiquitous dSesn expression in wild-type flies does not account for survival differences observed in flies expressing polyQ-expanded ATXN2.

while the D424A (dSesn$^{D424A}$) and D423A/D424A (dSesn$^{D423A/D424A}$) substitutions disrupt TORC1 inhibiting and TORC2/Akt potentiating functions of dSesn (*Kim et al., 2015*). We previously showed that interaction with mTOR is required for the mobility-extending effects of dSesn in wild-type, sedentary flies (*Kim et al., 2020*). Overexpression of either wild-type or mutated dSesns did not increase

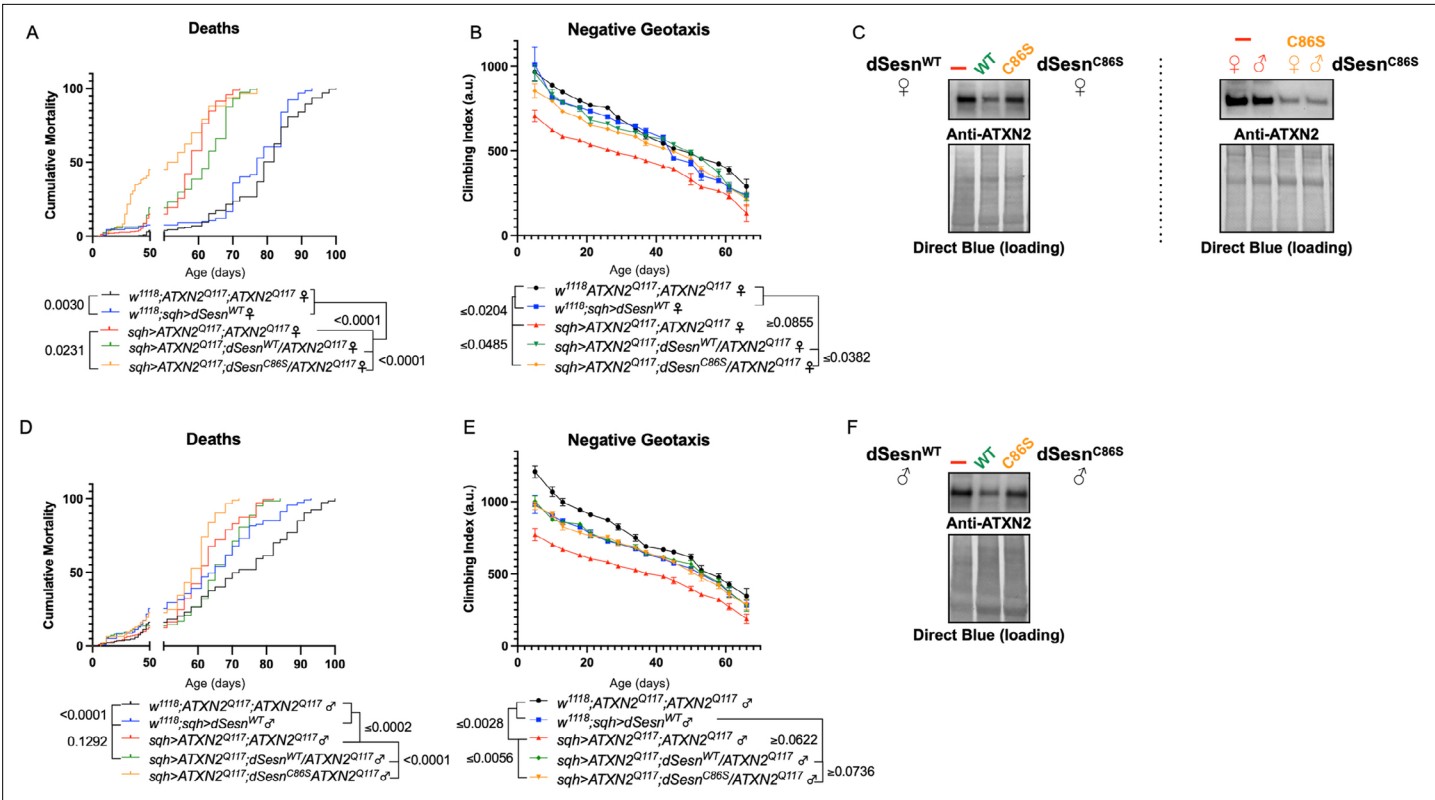

**Figure 5.** Oxidoreductase function is dispensable for mobility-extending effects of dSesn in flies expressing polyQ-expanded ATXN2. Ubiquitous expression of dSesn harboring a point mutation that abolishes oxidoreductase activity (*dSesn^C86S*, orange lines) exacerbates early death in SCA2 model flies (compare orange lines to red lines) in both females (**A**) and males (**D**). In contrast, climbing speed is partially rescued in the same female flies (expressing both dSesn^C86S and polyQ-expanded ATXN2, in orange) (**B**) and fully rescued in males (**E**). (**C, F**) Both wild-type dSesn and dSesn^C86S expression reduce ATXN2 levels, but the effect of dSesn^C86S expression is variable (examples of variability shown in panel [**C**]). Survival and mobility experiments performed in triplicate. Individual survival experiments are composed of ≥200 flies/genotype, scored every second day for death and analyzed by log-rank tests. Each individual climbing datum depicts average climbing speed for 5 vials of 20 flies, analyzed by 2-way ANOVA. Error bars indicate ± SD. Representative Western blots from five biological replicates (five flies per lysate), quantified in *Figure 7*.

The online version of this article includes the following source data for figure 5:

**Source data 1.** Uncropped, unedited blots from *Figure 5C* (females) and *Figure 5F* (males).

**Source data 2.** Uncropped, unedited blots from *Figure 5C* depicting examples of variability.

longevity in a wild-type background (i.e. without polyQ protein expression; *Figure 4—figure supplement 2*). To explore the requirement of Sesn's various functions for suppression of polyQ phenotypes, we performed mobility, survival, and protein expression experiments in which we ubiquitously overexpressed dSesn^C86S, dSesn^D424A, dSesn^D423A/D424A, or wild-type dSesn (dSesn^WT) in SCA2 flies.

First, we overexpressed dSesn without its antioxidant function(sqh>ATXN2Q; dSesnC86S;ATXN2Q117). Ubiquitous expression of dSesnC86S in SCA2 model flies did not provide any protection—in fact, it exacerbated premature death in both male and female flies (*Figure 5A, D*). However, we observed pronounced mobility improvements in female flies (*Figure 5B*) and mobility was fully rescued in males (*Figure 5E*). Whereas wild-type dSesn overexpression consistently reduced ATXN2 levels in both female (*Figure 5C*) and male (*Figure 5F*) SCA2 flies, the effect of dSesnC86S on ATXN2 levels was more variable (*Figure 5C, F*, quantified in Figure 7). These results suggest that while the oxidoreductase function of dSesn is 14 *Figure 6*: Interaction with mTOR is required for dSesn to improve survival and mobility inflies expressing polyQ-expanded ATXN2. Ubiquitous expression of dSesn harboring two separate mutations that abolish mTORC interaction dSesnD424A (A–D), dSesn D423A/D424A (E–H)(denoted by orange lines) exacerbates early death (A,E, females, C,G, males) and fails to rescuemobility (B,F, females, D,H, males) in SCA2 flies. Survival and mobility experiments performed intriplicate. Individual survival experiments are composed of ≥200 flies/genotype, scored

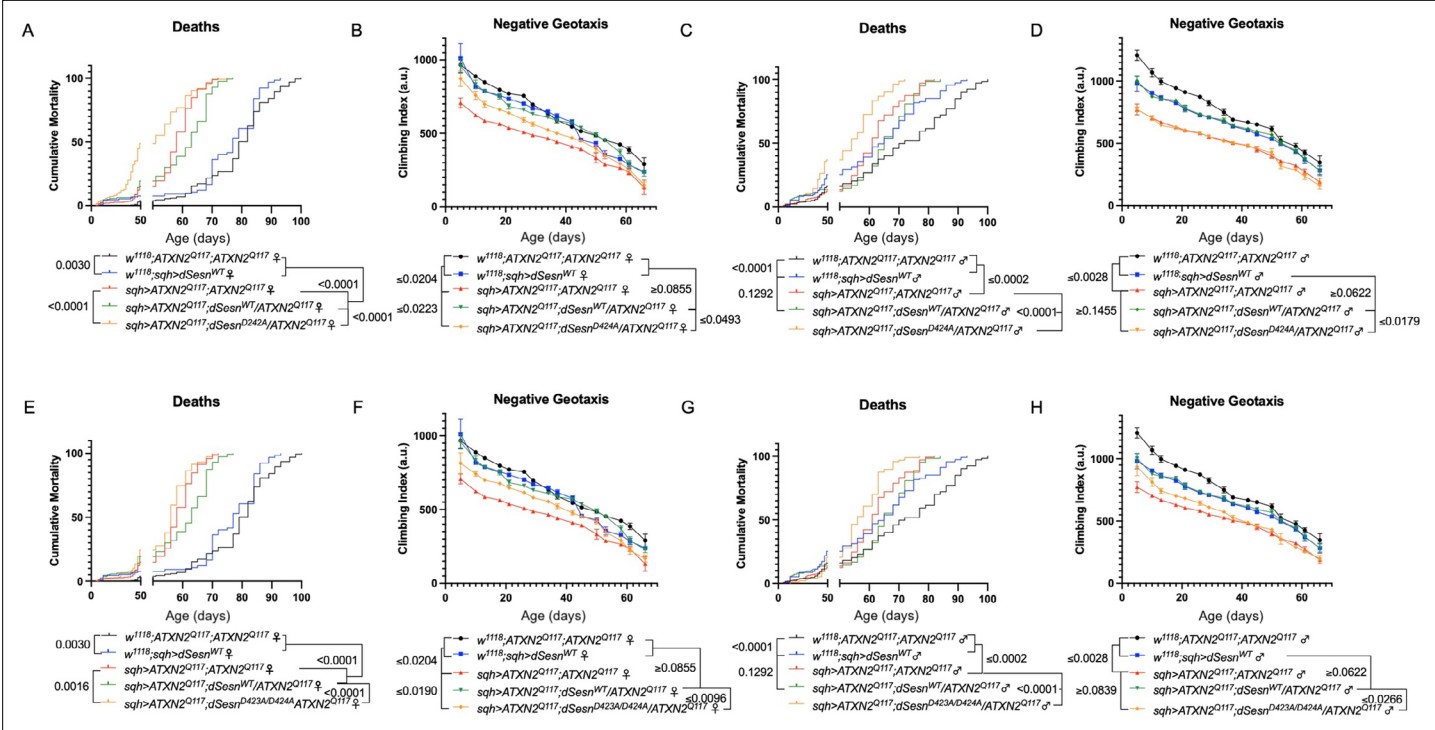

**Figure 6.** Interaction with mTOR is required for dSesn to improve survival and mobility in flies expressing polyQ-expanded ATXN2. Ubiquitous expression of dSesn harboring two separate mutations that abolish mTORC interaction dSesn$^{D424A}$ (**A–D**), dSesn $^{D423A/D424A}$ (**E–H**) (denoted by orange lines) exacerbates early death (A,E, females, C,G, males) and fails to rescue mobility (B,F, females, D,H, males) in SCA2 flies. Survival and mobility experiments performed in triplicate. Individual survival experiments are composed of ≥200 flies/genotype, scored every second day for death and analyzed by log-rank tests. Each individual climbing datum depicts average climbing speed for ≥5 vials of 20 flies, analyzed by 2-way ANOVA. Error bars indicate ± SD.

every second day for death and analyzed by log-rank tests. Each individual climbing datum depicts average climbing speed for ≥5 vials of 20 flies, analyzed by 2-way ANOVA. Error bars indicate ± SD not required for protection against motility defects in SCA2 flies, it is essential for improved survival.

We next ubiquitously expressed two separate amino-acid substitutions of dSesn that knock out its ability to modulate mTOR (**Kim et al., 2015**). Like dSesn$^{C86S}$ experiments, expression of either dSesn$^{D424A}$ (**Figure 6A–D**) or dSesn$^{D423A/D424A}$ (**Figure 6E–H**) increased early death in both female (**Figure 6A, E**) and male (**Figure 6C, G**) SCA2 flies. In female flies, both dSesn$^{D424A}$ and dSesn $^{D423A/D424A}$ conferred modest improvements to climbing speed compared to SCA2 model flies, but improvement did not equal that of wild-type dSesn expression and was absent at later timepoints (**Figure 6B, F**). Male flies ubiquitously expressing dSesn$^{D424A}$ did not display rescued mobility defects at any age (**Figure 6D**), while overexpression of dSesn $^{D423A/D424A}$ provided modest climbing speed improvements at early ages only (**Figure 6H**). Thus, interaction with mTOR appears critical for full rescue of climbing speed and protection against early death in SCA2 flies.

To examine the relative impact of dSesn expression on disease protein levels, we again performed Western blots on flies ubiquitously expressing polyQ-expanded ATXN2 with and without simultaneous expression of either wild-type dSesn, or dSesn harboring mutations that separately abolish its oxidative protection or mTOR modulating functions. ATXN2 protein in SCA2 model flies (*sqh>ATXN2$^{Q117}$;ATXN2$^{Q117}$*) was easily visible by log-phase of mortality (**Figure 7A–C, F**). All mutant dSesn-expressing cohorts had variable, but statistically significant reductions in ATXN2 protein levels.

Based on these results and on our prior observations in exercised flies and flies with muscle-specific dSesn overexpression (**Kim et al., 2020**; **Sujkowski et al., 2020**), we hypothesized that dSesn overexpression and reduced ATXN2 levels relate to changes in autophagy. We observed that AtgIIa/Ia ratios increased in both male and female SCA2 model flies that were overexpressing wild-type dSesn, while its mutated versions had higher variability, which largely trended towards increased autophagy, but

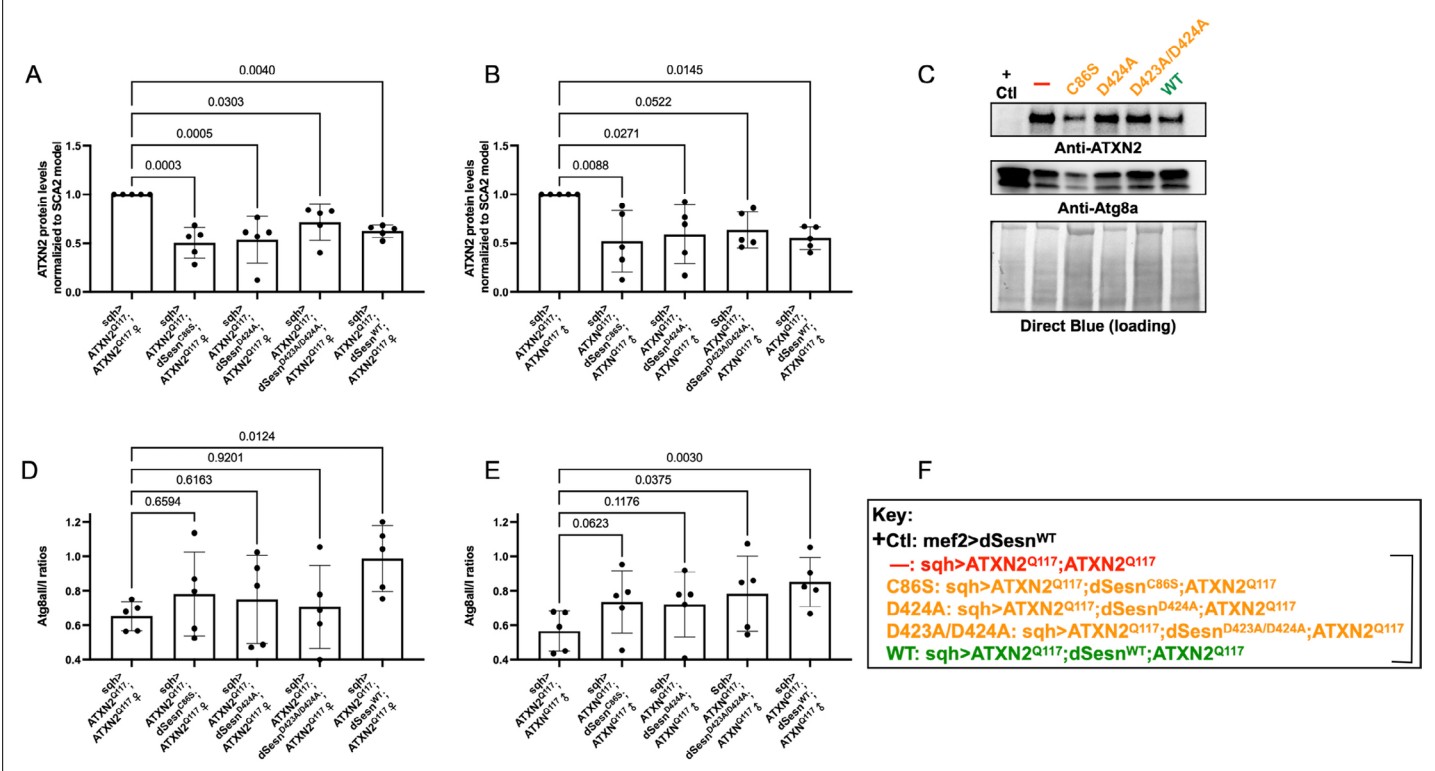

**Figure 7.** Wild-type dSesn expression reduces disease protein and increases autophagy in flies expressing polyQ-expanded ATXN2. Ubiquitous expression of dSesn significantly reduces ATXN2 levels (**A–C**) and increases AtgIIa;AtgIa ratios (**C–E**). $n$ = 5 biological replicates, 5 flies per lysate, analyzed by ANOVA with Tukey post-hoc comparison for significantly different groups. (**F**) Genotypes in representative Western blot and from independent biological replicates, quantified in (**A, B, D, E**). Bracket in (**F**) indicates quantifications in (**A, B, D, E**).

The online version of this article includes the following source data for figure 7:

**Source data 1.** Uncropped, unedited blots from *Figure 7C*.

without reaching significance (*Figure 7C–F*). These data may reflect different mechanisms of protection at play, discussed further below.

Finally, since dSesn overexpression improved survival in flies ubiquitously expressing two copies of the polyQ-expanded ATXN2 transgene (*Figure 4A, D*), we repeated our longevity experiments in exercised SCA2 flies expressing two copies of the ATXN2 transgene. Exercised female flies expressing two copies of CAG-expanded *ATXN2* either ubiquitously (*Figure 8A*) or in adult neurons (*Figure 8C*) significantly improved survival compared to unexercised siblings. Male SCA2 flies trended toward improved survival in early life only, while they were still undergoing exercise training (*Figure 8B, D*). Altogether, these findings indicate protective effects from endurance exercise not only in terms of motility (*Figures 1 and 2*) but also in terms of longevity (*Figure 8*) in SCA2 flies.

In summary, both exercise and *dSesn* expression ameliorated mobility and survival defects in SCA2 model flies and these improvements correlated with reduced disease protein levels. Furthermore, these neuroprotective effects relied on known functions of dSesn related to mTOR and oxidoreductase activities (*Figure 9*).

## Discussion

Endurance exercise enhances quality of life and provides remedy for a number of conditions and diseases, but, until now, has not been systematically examined across several polyQ disorders. Lack of systematic studies in this respect also means that exercise-dependent molecular mechanisms that can be utilized to fight polyQ diseases remain largely unknown. Here, we explored in a highly controlled and rigorously quantified way the possibility that daily training is of benefit against some polyQ

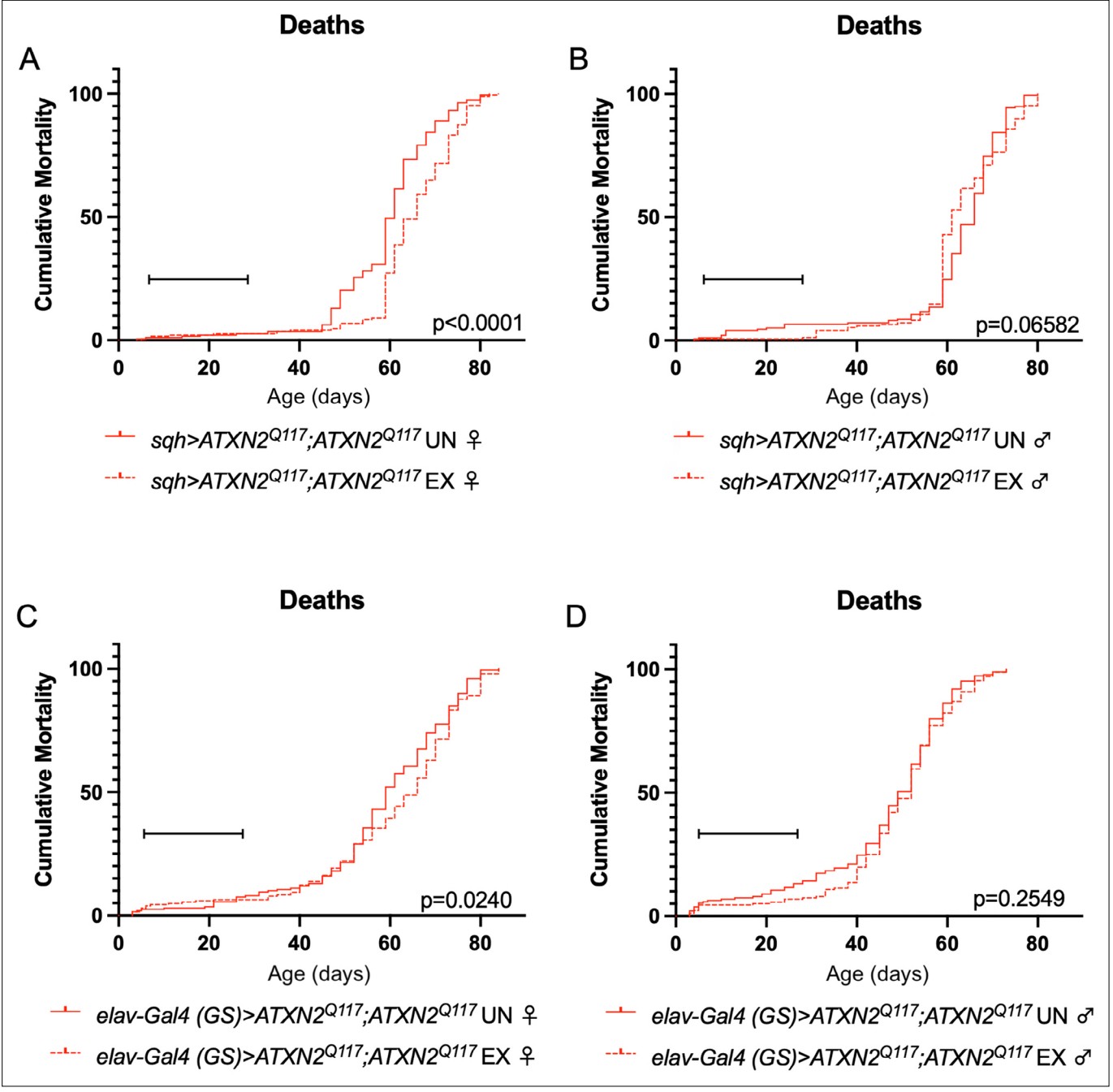

**Figure 8.** Exercise reduces early death in SCA2 model flies expressing two copies of *ATXN2*. Exercised female flies expressing two copies of CAG-expanded *ATXN2* ubiquitously (**A**) or in adult neurons (**C**) have increased survival compared to age-matched, unexercised siblings, while exercised male flies expressing two copies of CAG-expanded *ATXN2* ubiquitously (**B**) or in adult neurons (**D**) trend toward increased survival only in the first 25 days, the period in which flies are still training. Brackets indicate exercise training period. p-Values indicate log-rank for entire survival curve, $n \geq 170$, performed in duplicate.

diseases. In addition, we identified a key factor of exercise's protective effects that can be harvested towards therapeutic options.

Endurance exercise significantly rescued progressive motility defects in SCA2 flies, allowing their climbing speed and endurance to remain in wild-type range. Exercise also partially rescued SCA6 flies, with no beneficial effect on SCA3 model flies. Thus, exercise can have potent rescue effects for particular polyQ diseases. To obtain early insight into how exercise could lead to such dramatic improvement in SCA2 flies (albeit not for SCA3) we examined protein levels of ATXN2 and ATXN3,

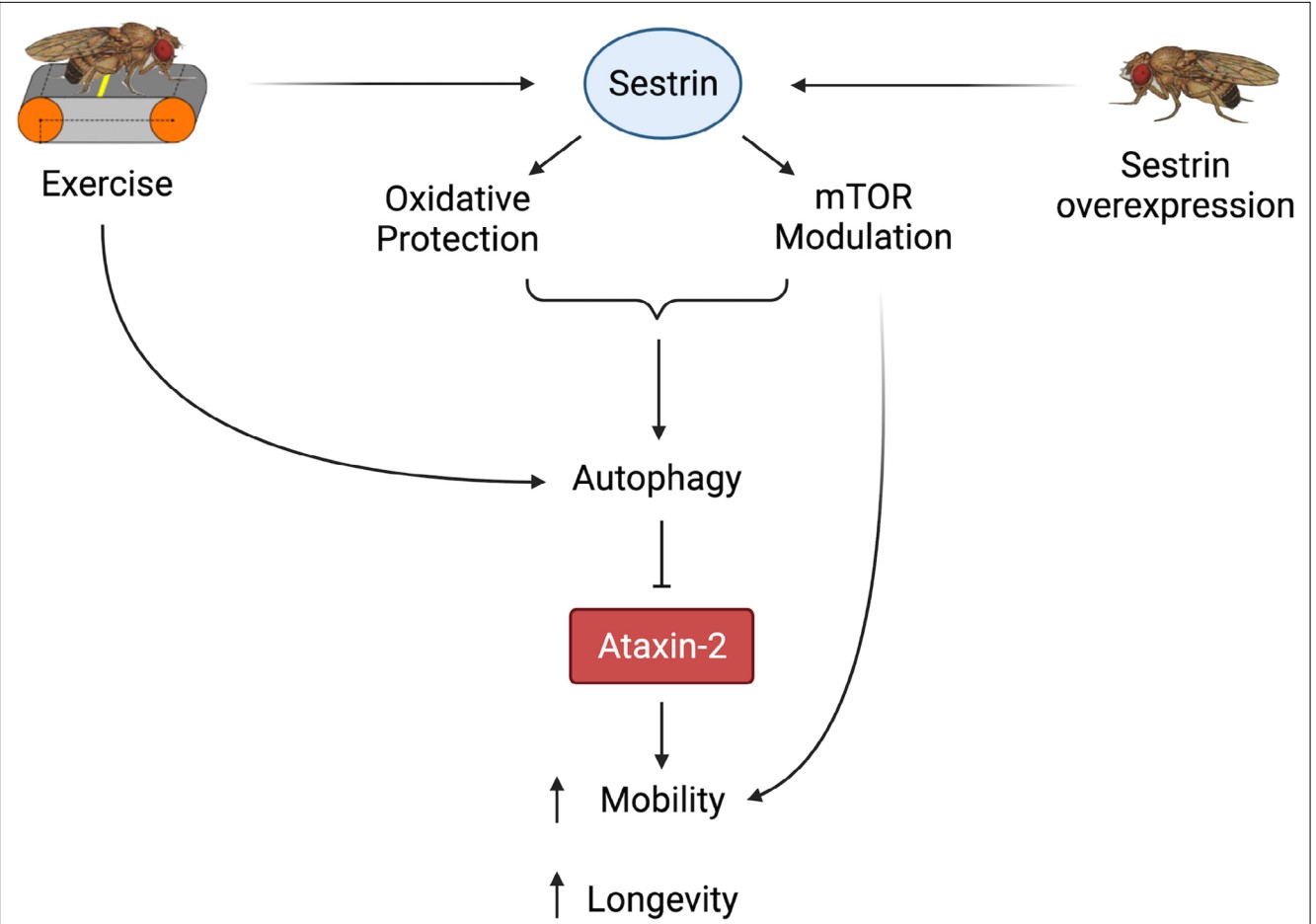

**Figure 9.** Proposed model of the effects of exercise and dSesn overexpression on SCA2 flies. Sesn's known functions and activation by exercise have been established previously (*Kim et al., 2020*; *Kim et al., 2015*). Wild-type Sesn overexpression or exercise can activate autophagy and reduce disease protein levels, improving function. Source data-all figures. Tab delimited excel file containing raw data for all figures and figure supplements. Uncropped, unedited Western blot data are included as separate images.

their respective disease proteins. Exercise led to reduction in protein levels of ATXN2, but did not noticeably affect ATXN3. In the context of endurance exercise, improved motility in SCA2 flies may be due to reduced disease protein levels.

In addition, we examined the effect of the powerful exercise-mimetic protein, Sesn, on lifespan, motility, and disease protein levels in SCA2 model flies. Sesns are induced by oxidative and genotoxic stress in flies and mammals and promote autophagy (*Lee et al., 2010*; *Budanov and Karin, 2008*). Sesns protect against oxidative stress-induced fly neuronal death (*Singh and Chowdhuri, 2018*) and reduce accumulation of protein aggregates in mammalian cells (*Reddy et al., 2016*). In previous work with wild-type flies, we found that expression of fly *Sesn* is sufficient to preserve speed and endurance and has no additive effect with exercise, consistent with a role as a key effector of exercise (*Kim et al., 2020*). Here, we found that Sesn was protective in SCA2 model flies, but did not confer benefit to either SCA3 or SCA6. Furthermore, Sesn's oxidoreductase function was not necessary to rescue mobility defects in SCA2, while both its oxidative resistance and mTOR modulating activities were required for Sesn to improve early death. In fact, overexpression of Sesn with mutations eliminating either its oxidoreductase activity or mTOR modulating functions exacerbated premature death in SCA2 flies.

We have previously shown that the oxidoreductase activity of Sesn is dispensable for motility improvements in wild-type flies (*Kim et al., 2020*), similar to our observations with SCA2 flies in this study. Sesn overexpression is also not sufficient to extend lifespan in a wild-type background, shown here and elsewhere (*Kim et al., 2020*; *Lee et al., 2010*). Neurons are particularly susceptible

to oxidative stress and reactive oxygen species can induce cell death during neurodegeneration (*Fatokun et al., 2008*). Perhaps, in the context of neurodegenerative disease, Sesn protects mobility until oxidative burden in neurons becomes insurmountable.

Wild-type Sesn overexpression in SCA2 model flies correlated with significantly reduced disease protein and increased autophagic flux. Overexpression of mutated versions of Sesn also led to reduced disease protein levels. This is not entirely surprising: mutated Sesns still had some positive impact on SCA2 fly mobility, but did not have the same, fully protective effect of wild-type Sesn. Collectively, these findings suggest the possibility of separate mechanisms at play: some protective mechanisms may directly depend on reduced toxic protein levels (as observed before in a model of HD and exercise *Mangiarini et al., 1996*; *van Dellen et al., 2008*); others may protect independently of changes in degenerative protein levels, or perhaps without directly relying on it. Future investigations are required to parse out such mechanisms and whether they function in a cell autonomous or non-cell autonomous manner.

To conclude, we propose endurance exercise as a promising therapeutic intervention for polyQ neurodegeneration, particularly in SCA2. We also demonstrate that Sesn can substitute for exercise benefits in flies that do not exercise. Identification of molecular targets capable of inducing exercise-like enhancements in the presence of neurodegeneration is particularly important, as many patients will eventually lose the ability to exercise. This work has begun to uncover exercise-induced mechanisms that can be utilized for neuroprotection against specific polyQ diseases and can be leveraged toward therapeutics in the future.

# Materials and methods

### Key resources table

| Reagent type (species) or resource | Designation | Source or reference | Identifiers | Additional information |
|---|---|---|---|---|
| Gene (*H. sapiens*) | ATXN2 | GenBank | FLYB: FBgn0267931 | |
| Gene (*H. sapiens*) | ATXN3 | GenBank | FLYB: FBgn0024961 | |
| Gene (*H. sapiens*) | CACNA1A | GenBank | FLYB: FBgn0283733 | |
| Gene (*D. melanogaster*) | Sesn | GenBank | FLYB: FBgn0034897 | |
| Genetic reagent (*D. melanogaster*) | UAS- ATXN2$^{Q117}$ | Bloomington *Drosophila* Stock Center | BDSC68394: FBst0068394 | Flybase Symbol:w$^*$; P{UAS-ATXN2.117Q}8B |
| Genetic reagent (*D. melanogaster*) | UAS- ATXN2$^{Q117}$ | Bloomington *Drosophila* Stock Center | BDSC68395: FBst0068395 | Flybase Symbol: w$^*$; P{UAS-ATXN2.117Q}9 A |
| Genetic reagent (*D. melanogaster*) | UAS-ATXN3$^{Q77}$ | *Tsou et al., 2016* | | |
| Genetic reagent (*D. melanogaster*) | UAS-α1ACT$^{Q70}$ | *Sutton et al., 2017* | | |
| Genetic reagent (*D. melanogaster*) | w1118;UAS-ATXN2$^{Q117}$/CyO;UAS-ATXN2$^{Q117}$ | This paper | | See methods lines 305–306 |
| Genetic reagent (*D. melanogaster*) | w1118;sqh/CyO;UAS- dSesn$^{WT}$/TM3-Sb | This paper | | See methods lines 311–314 |
| Genetic reagent (*D. melanogaster*) | w1118;sqh/CyO;UAS- dSesn$^{C86S}$/TM3-Sb | This paper | | See methods lines 311–314 |
| Genetic reagent (*D. melanogaster*) | w1118;sqh/CyO;UAS- dSesn$^{D424A}$/TM3-Sb | This paper | | See methods lines 311–314 |
| Genetic reagent (*D. melanogaster*) | w1118;sqh/CyO;UAS- dSesn$^{D423A/D424A}$/TM3-Sb | This paper | | See methods lines 311–314 |
| Antibody | anti-ataxin-2 (mouse monoclonal) | BD biosciences | 611,378 | (1:500) |
| Antibody | anti-ataxin-3 (mouse monoclonal) | Millipore | 1H9, MAB5360 | (1:1000) |
| Antibody | anti-GABARAP (rabbit polyclonal) | Abcam | Ab1398 | (1:1000) |
| Antibody | anti-dSesn (rabbit polyclonal) | *Kim et al., 2020* | | (1:500) |

## Antibodies

Primary antibodies were obtained from the following sources: anti-ATXN2 (mouse monoclonal, 1:500; BD biosciences, Franklin Lakes, NJ) anti-ATXN3 (mouse monoclonal 1H9, MAB5360, 1:500–1000; Millipore, Burlington, MA), anti-dSesn (rabbit polyclonal, 1:500, *Lee et al., 2010*), anti-GABARAP (rabbit polyclonal, 1:1000, Abcam, Cambridge, United Kingdom). Peroxidase-conjugated secondary antibodies (goat anti-mouse, goat anti-rabbit, 1:5000; Jackson Immunoresearch, West Grove, PA).

## Fly stocks and maintenance

Gifted stocks used in this study were sqh-Gal4 (Daniel Kiehart, Duke University, abbreviated "sqh" throughout), and $w^{1118}$ (Russ Finley, Wayne State University). UAS-Q80, UAS-ATXN3$^{Q77}$, UAS-α1ACT$^{Q70}$ and elav-Gal4 (GS) were previously described in *Tsou et al., 2016*; *Johnson et al., 2020*; *Johnson et al., 2019* UAS- ATXN2$^{Q117}$ (BDSC#68394, BDSC#68395), on chromosomes II and III, respectively, were obtained from the Bloomington *Drosophila* Stock Center (Bloomington, IN). Both SCA2 lines were used for pilot work. A single line containing UAS-ATXN2$^{Q117}$ on both chromosomes II and III was then generated using standard crosses (w1118;UAS-ATXN2$^{Q117}$/CyO;UAS-ATXN2$^{Q117}$).

Wild-type (dSesn$^{WT}$) and C86S(dSesn$^{C86S}$), D424A (dSesn$^{D424A}$) or D423A/D424A (dSesn$^{D423A/D424A}$) mutations are described in *Kim et al., 2020*; *Kim et al., 2015*. C86, D423 and D424 in dSesn correspond to C125, D406 and D407 in mammalian SESN2 (*Kim et al., 2015*). These four lines were separately recombined with sqh-Gal4 to create four independent lines with genotypes: w1118;sqh/ CyO;UAS- dSesn$^{XX}$/TM3-Sb. Simultaneous overexpression of both UAS-ATXN2$^{Q117}$ and UAS-dSesn$^{XX}$ was achieved via a single cross between w1118; UAS-ATXN2$^{Q117}$/CyO;UAS-ATXN2$^{Q117}$ virgin females and w1118;sqh/CyO;UAS-dSesn$^{XX}$/TM3-Sb males to generate w1118;UAS-ATXN2$^{Q117}$;UAS-dSesn$^{XX}$/ ATXN2$^{Q117}$. All insertions were validated by Western blotting (*Tsou et al., 2015b*; *Tsou et al., 2016*; *Ristic et al., 2018*; *Johnson et al., 2020*; *Lee et al., 2010*; *Johnson et al., 2019*; *Blount et al., 2018*).

Prior to all experiments, fly cultures were maintained at a constant density for at least two generations. About 20–25 virgin females (depending on genotype) and 5 males were mated in 300 mL bottles with 50 mL standard 10% sucrose 10% yeast spiked with 500 µL Penicillin-Streptomycin 10,000 u/mL, 10 mg/mL in 0.9% sterile NaCl (Sigma-Aldrich, St. Louis, MO). Adult progeny were synchronized by collecting within 6 hr of eclosion over a 24 hr time period. Groups of 20 age- and sex-matched flies were immediately transferred into narrow polypropylene vials containing 5 mL of standard 10% sucrose 10% yeast (no antibiotic) or RU486 food, as appropriate. Food vials were changed every day during exercise and every second day thereafter and scored for mobility and lifespan until no flies remained.

Flies were housed in a 25°C incubator on a 12:12 h light:dark cycle at 50% relative humidity. Control flies for all non-gene-switch Gal4 UAS experiments consisted of both the UAS and Gal4 lines into $w^{1118}$. For gene-switch experiments, RU- flies of the same genotype served as the negative control. RU+ group received 100 µM RU486/mifepristone (Cayman Chemical, Ann Arbor, MI), which activates the gene switch (GS) driver, while RU- group received the same volume of vehicle solution (70% ethanol).

## Exercise training

Triplicate cohorts of at least 800 flies were collected under light $CO_2$ anesthesia within 6 hr of eclosion and separated into vials of 20. Flies were then further separated into two large cohorts of at least 400 flies, which served as exercised and unexercised groups. Exercised groups received three weeks of ramped exercise as described previously (*Piazza et al., 2009*). The unexercised groups were placed on the exercise training device at the same time as the exercised groups, but were prevented from running by the placement of a foam stopper low in the vial. Both cohorts were housed in the same incubator with normal foam stopper placement at all times other than during an exercise bout.

## Climbing speed

Negative geotaxis was assessed in Rapid Negative Geotaxis (RING) assays in groups of at least 100 flies as described (*Damschroder et al., 2018a*). Briefly, vials of 20 flies were briskly tapped down, then measured for climbing distance after 2 s of inducing the negative geotaxis instinct. For each group of vials, an average of five consecutive trials was calculated and batch-processed using ImageJ (Bethesda, MD). Flies were longitudinally tested 2–3 times per week for 5–7 weeks. Between assessments, flies were returned to food vials and housed normally as described above. Negative geotaxis

results were analyzed using two-way ANOVA analysis (age-effect and genotype effect) with post hoc Tukey's multiple comparison tests in GraphPad Prism (San Diego, CA). All negative geotaxis experiments were performed in triplicate, with one complete trial shown in each graph.

### Endurance

At least eight vials of 20 flies from each cohort were subjected to the endurance analysis on day 25 of adulthood, immediately after exercise was complete and after 2 days of recovery. For each session, the flies were placed on the Power Tower exercise machine (*Sujkowski and Wessells, 2018*) and the climbing instinct was induced until flies no longer responded to the negative geotaxis stimulus. Monitored at 15 min intervals, a vial of flies was visually determined to be fatigued when 10% or fewer flies could climb higher than 1 cm after three consecutive drops. Each vial was plotted as a single datum. Endurance experiments were performed in triplicate and at the same time as age-matched background controls and were scored blindly wherever possible. The time from the start of the assay to the time of fatigue was recorded for each vial, and the data analyzed using log-rank analysis in GraphPad Prism (San Diego, CA). Each graph represents an independent, representative repetition.

### Longevity

For exercise experiments, appropriate food vials (RU486, vehicle, or standard 10% sucrose 10% yeast) were changed and deaths were recorded five times per week during exercise training and three times per week after completion of the training program. All other longevity experiments were scored three times per week. Dead flies were removed and counted until no flies remained. Differences in survival were plotted as cumulative mortality and assessed following censoring using log-rank analysis in GraphPad Prism (San Diego, CA). Longevity experiments were performed in triplicate and in parallel with background controls, with each individual graph depicting a representative biological repetition.

### Western blots

Three-to-five whole flies per biological replicate, depending on experiment, were homogenized in boiling lysis buffer (50 mM Tris pH 6.8, 2% SDS, 10% glycerol, 100 mM dithiothreitol), sonicated, boiled for 10 min, and centrifuged at 13,300xg at room temperature for 10 min. Western blots were developed ChemiDoc (Bio-Rad, Hercules, CA) and quantified with ImageLab (Bio-Rad, Hercules, CA). For direct blue staining, PVDF membranes were submerged for 10 min in 0.008% Direct Blue 71 (Sigma-Aldrich, St. Louis, MO) in 40% ethanol and 10% acetic acid, rinsed in 40% ethanol/10% acetic acid, air dried, and imaged. Western blots were performed using at least three biological replicates, and statistical analysis was performed in GraphPad Prism (San Diego, CA).

### Statistical analysis

Survival and endurance analyses were analyzed by log-rank. Non-natural deaths were censored and did not exceed 10% of flies for any group. Climbing speed over time was analyzed by two-way ANOVA for age effect and genotype effect, with Tukey post-hoc comparison between significantly different groups. Western blots were analyzed using either student *t*-test or ANOVA depending on experiment. All statistics were performed using GraphPad Prism software version 9.2.0 (San Diego, CA, USA) for Macintosh.

## Acknowledgements

We thank Henry Paulson for the anti-MJD antibody, and Todi lab members Kozeta Libohova, Wei-Ling Tsou, Gorica Ristic, Joanna Sutton, Jessica Blount, and Sean Johnson for generation and validation of the SCA lines used in this study. We acknowledge our collaborators Myungjin Kim and Jun Hee Lee for the original dSesn *Drosophila* stocks and reagents. This research was funded by the Wayne State University Thomas C Rumble Graduate Fellowship (to ALS), NIH R01 AG059683 (RJW), NIH R21 NS121276 (RJW and SVT), NIH R01 NS086778 (SVT), and NIH/NHLBI 2T32HL120822 (KR).

# Additional information

## Funding

| Funder | Grant reference number | Author |
| --- | --- | --- |
| Wayne State University | Thomas C. Rumble Graduate Fellowship | Alyson Sujkowski |
| NIH Office of the Director | R01 AG059683 | Robert J Wessells |
| NIH Office of the Director | R21 NS121276 | Robert J Wessells Sokol V Todi |
| NIH Office of the Director | R01 NS086778 | Sokol V Todi |
| National Heart, Lung, and Blood Institute | 2T32HL120822 | Kristin Richardson |

The funders had no role in study design, data collection and interpretation, or the decision to submit the work for publication.

## Author contributions

Alyson Sujkowski, Conceptualization, Data curation, Formal analysis, Funding acquisition, Supervision, Visualization, Writing - original draft, Writing – review and editing; Kristin Richardson, Matthew V Prifti, Data curation, Writing – review and editing; Robert J Wessells, Sokol V Todi, Conceptualization, Funding acquisition, Methodology, Supervision, Writing – review and editing

## Author ORCIDs

Alyson Sujkowski ![ORCID] http://orcid.org/0000-0002-9909-9279
Robert J Wessells ![ORCID] http://orcid.org/0000-0002-4516-3183
Sokol V Todi ![ORCID] http://orcid.org/0000-0003-4399-5549

## Decision letter and Author response

Decision letter https://doi.org/10.7554/eLife.75389.sa1
Author response https://doi.org/10.7554/eLife.75389.sa2

# Additional files

## Supplementary files
- Transparent reporting form
- Source data 1. *Figures 1–8* source data.

## Data availability

All data generated or analysed during this study are included in the manuscript and supporting file; source data files have been provided for all figures and supplementary information.

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
