## [Editor Report]

In this strong study, the investigators examine the therapeutic ability of endurance exercise to disease in *Drosophila* models of the spinocerebellar ataxias types (SCAs) 2, 3, and 6. Results support that exercise induced improvements rely on Sesn and perhaps its role in regulating autophagy.

---

## [Decision Letter]

**Decision letter after peer review:**

Thank you for submitting your article "Endurance exercise ameliorates phenotypes in *Drosophila* models of Spinocerebellar Ataxias" for consideration by *eLife*. Your article has been reviewed by 3 peer reviewers, including Harry T Orr as the Reviewing Editor and Reviewer #1, and the evaluation has been overseen by K VijayRaghavan as the Senior Editor. The following individual involved in review of your submission has agreed to reveal their identity:; Amit Singh (Reviewer #3).

Summary:

The ability of exercise to mitigate the disease-associated phenotypes for the inherited ataxias is an area under investigation for several years. Previous studies explore the ability of endurance exercise to promote neuronal plasticity and memory as well as diminish neurodegeneration. Here the investigators examine the therapeutic ability of endurance exercise to disease in *Drosophila* models of the spinocerebellar ataxias types (SCAs) 2, 3, and 6. Overall, this is a very strong manuscript.

Essential revisions:

1. Throughout the manuscript the wrong figures are referred to in the text. Examples include Figure 1C instead of Figure 2C in line 135 and on page 7 Figure 3 instead of Figure 4 is written more than once.

2. In almost every figure legend is a lack of information on number of flies or samples per group and the exact statistical comparisons and post-hoc tests used. It is far more useful to have this information directly in the figure legend, especially for readers who go straight to the figures and do not focus on the text as much. For example, Figure 2 states N>100, n>8 vials of 20 flies, but do the data points represent the mean of each vial's average, or N>100 as an entire group? What statistical test is used? The same is true for Figure 3-6, and additionally these figures lack information on how many vials were in each group (i.e. are there any batch effects from vials that could skew the results?).

3. The data for Figure 7 is the key supporting evidence of the claim that this is operating via autophagy. However, given the variability in the data and the very small sample sizes N=3-4/group, this is not currently supported. For example, the effects of the dSEN WT expression on ATXN2 protein levels is only significant because the grouping is tighter but there is a decrease in the other dSesn mutants. Same is true for AtgIa/IIa ratios where there is some effect of the dSesn mutant expression but just more variability. Either more samples are needed, or these data and claims should be removed from the manuscript.

Additional points:

1. It is stated that Sesn administered to SCA3 and SCA6 flies was of no benefit, lines 174, 175 and 261 but the data are not shown. These data need to included in supplemental information.

2. Quantification of the data for Figure 1 should be added to that figure.

3. Rename the C86S mutant as dSesn^C86S as it is confusing to the reader to shorten to dSesn^CS.

4. Describe the phenotype of SCA2 expression in Sestrin mutant background.

5. How do the two domains of Sestrin trigger autophagic flux response?

6. Is there a role of caspase dependent cell death in SCA2 neurodegenerative phenotype?

---

## [Author Response]

Essential revisions:1. Throughout the manuscript the wrong figures are referred to in the text. Examples include Figure 1C instead of Figure 2C in line 135 and on page 7 Figure 3 instead of Figure 4 is written more than once.

Thank you for noticing these errors, and we apologize for them. Figure callouts have been corrected and now appear in lines 133 and 166-170.

2. In almost every figure legend is a lack of information on number of flies or samples per group and the exact statistical comparisons and post-hoc tests used. It is far more useful to have this information directly in the figure legend, especially for readers who go straight to the figures and do not focus on the text as much. For example, Figure 2 states N>100, n>8 vials of 20 flies, but do the data points represent the mean of each vial's average, or N>100 as an entire group? What statistical test is used? The same is true for Figure 3-6, and additionally these figures lack information on how many vials were in each group (i.e. are there any batch effects from vials that could skew the results?).

We appreciate this suggestion which greatly improves the accessibility of the manuscript. Clarifications have been made directly in figure legends 2-6, and again in lines 587-591, 603-607, 625-631, 642, 651-655, 661-663, 674, 683-688, 699-703, 707-710, and 719-720 to fully address these recommendations.

3. The data for Figure 7 is the key supporting evidence of the claim that this is operating via autophagy. However, given the variability in the data and the very small sample sizes N=3-4/group, this is not currently supported. For example, the effects of the dSEN WT expression on ATXN2 protein levels is only significant because the grouping is tighter but there is a decrease in the other dSesn mutants. Same is true for AtgIa/IIa ratios where there is some effect of the dSesn mutant expression but just more variability. Either more samples are needed, or these data and claims should be removed from the manuscript.

We agree that these claims can be strengthened with additional data, which have now been added to Figure 7. Western blots here are now representative of 5 biological replicates consisting of lysates of 5 individual flies/genotype. Quantifications are analyzed by ANOVA with Tukey post-hoc comparison for significantly different groups. Results are discussed in lines 214-222, pasted below:

“All mutant dSesn-expressing cohorts had variable, but statistically significant reductions in ATXN2 protein levels.

Based on these results and on our prior observations in exercised flies and flies with muscle-specific dSesn overexpression [40, 64], we hypothesized that dSesn overexpression and reduced ATXN2 levels relate to changes in autophagy. We observed that AtgIIa/Ia ratios increased in both male and female SCA2 model flies that were overexpressing wild-type dSesn, while its mutated versions had higher variability which largely trended towards increased autophagy, but without reaching significance (Figure 7C, D-F).”

Additional points:1. It is stated that Sesn administered to SCA3 and SCA6 flies was of no benefit, lines 174, 175 and 261 but the data are not shown. These data need to included in supplemental information.

Thank you for this recommendation. These data are now included as Figure 4—figure supplement 1, referenced in line 173.

2. Quantification of the data for Figure 1 should be added to that figure.

Quantification is now added to Figure 1.

3. Rename the C86S mutant as dSesn^C86S as it is confusing to the reader to shorten to dSesn^CS.

Thank you for this suggestion. This has been changed throughout.

4. Describe the phenotype of SCA2 expression in Sestrin mutant background.

We have previously established that Sestrin mutants do not adapt to exercise. Relevant additional text is pasted below, found in lines 147-151.

“We have previously shown that the stress-inducible protein Sesn is necessary for exercise adaptations in multiple species (40), and Sesn activity has been proposed to play a protective role against neurodegeneration (57).”

5. How do the two domains of Sestrin trigger autophagic flux response?

We apologize for this omission. Clarification is found in lines 149-151, pasted below:

“Both endurance exercise and Sesn increase autophagy by inhibiting mTOR (33, 58, 59), and increased autophagy increases ATXN2 solubility and reduces phenotypes in SCA2 patient cells (60).”

And in the results lines 177-180:

“Sestrin is a multi-functional protein with separate, previously identified oxidoreductase and mTOR interacting domains (63). The C86S (dSesn^C86S^) mutation abolishes oxidoreductase activity while the D424A (dSesn^D424A^) and D423A/D424A (dSesn^D423A/D424A^) substitutions disrupt TORC1 inhibiting and TORC2/Akt potentiating functions of dSesn (63).”

And in the methods, lines 307-309

“Wild-type (dSesn^WT^) and C86S(dSesn^C86S^), D424A (dSesn^D424A^) or D423A/D424A (dSesn^D423A/D424A^) mutations are described in (40, 63). C86, D423 and D424 in dSesn correspond to amino acid residues C125, D406 and D407 in mammalian SESN2 (63).”

6. Is there a role of caspase dependent cell death in SCA2 neurodegenerative phenotype?

This is an interesting idea which we have not tested. We do have unpublished observations that indicate no apparent difference in ubiquitination patterns between exercised and unexercised SCA2 model flies, or SCA2 flies expressing dSesn. Although the ubiquitin-proteasome system plays a role in apoptosis, we have not yet identified a direct link here.